# Circular RNAs: Novel Players in Cancer Mechanisms and Therapeutic Strategies

**DOI:** 10.3390/ijms251810121

**Published:** 2024-09-20

**Authors:** Jimi Kim

**Affiliations:** 1Department of Life Sciences, Gachon University, Seongnam 13120, Republic of Korea; zimic@gachon.ac.kr; 2Department of Health Science and Technology, GAIHST, Lee Gil Ya Cancer and Diabetes Institute, Incheon 21999, Republic of Korea

**Keywords:** circular RNA, cancer biomarker, RNA therapeutics, circRNA vaccine

## Abstract

Circular RNAs (circRNAs) are a novel class of noncoding RNAs that have emerged as pivotal players in gene regulation. Our understanding of circRNAs has greatly expanded over the last decade, with studies elucidating their biology and exploring their therapeutic applications. In this review, we provide an overview of the current understanding of circRNA biogenesis, outline their mechanisms of action in cancer, and assess their clinical potential as biomarkers. Furthermore, we discuss circRNAs as a potential therapeutic strategy, including recent advances in circRNA production and translation, along with proof-of-concept preclinical studies of cancer vaccines.

## 1. Overview of Circular RNAs (circRNAs)

Circular RNAs (circRNAs) are single-stranded, covalently closed RNA molecules formed through non-canonical splicing events, specifically back-splicing. This unique structural feature renders them resistant to exonucleases, which makes them more stable than linear mRNAs.

Initially perceived as rare products of mis-splicing [1] or exon scrambling [2], circRNAs are widely expressed in animals [3]. Systematic sequencing and bioinformatic studies have led to an explosion in research on circRNAs, significantly expanding the catalog of circRNAs identified in humans [4,5,6].

Although many circRNAs exhibit low expression levels, certain species are highly abundant in specific tissues, notably the brain and testes, with expression patterns linked to developmental stages and physiological conditions [3,7,8]. Increasing evidence has implicated dysregulated circRNAs in the development, maintenance, and metastasis of cancers. This association was initially proposed with the discovery of ciRS-7 (also known as CDR1), which harbors multiple binding sites of miR-7 [6,9]. MiR-7 has been implicated in cancer development and epithelial–mesenchymal transition (EMT), suggesting ciRS-7′s potential role as a miR-7 inhibitor in cancer-related pathways [10]. Subsequent studies have demonstrated that several circRNAs are aberrantly regulated, closely associated with various cancers, and serve as potential biomarkers for diagnosis and prognosis. 

As our understanding of endogenous circRNAs grows, circRNAs have emerged as promising therapeutic platforms. Efforts are being made to develop efficient methods for effectively synthesizing circRNAs and producing proteins. Several companies and organizations are actively exploring the potential of circRNAs, including Sail Biomedicine (emerged from the merger of two flagship pioneering companies, Laronde and Senda Biosciences and focused on developing a new class of medicines based on endless RNA (eRNA) technology), Orna Therapeutics (developing circular RNA (oRNA) therapies), and Houston Methodist Research Institute (partnered with the Coalition for Epidemic Preparedness Innovations (CEPI) and exploring promising new circRNA vaccines) [11,12]. Continuous investment and research in this field highlight the significant potential of circRNAs in revolutionizing RNA-based medicine.

In this review, we provide an overview of the current understanding of circRNA biogenesis and explore their mechanisms of action and potential as biomarkers with selected examples in cancers. In addition, we discuss the recent updates on circRNA production and translation for the application of circRNAs.

## 2. Mechanisms of circRNA Formation

### 2.1. Biogenesis of circRNAs

CircRNA biogenesis is facilitated by back-splicing events that utilize canonical splice sites but employ distinct molecular mechanisms. In classical pre-mRNA splicing, the spliceosome recognizes the splicing sites, removes introns, and covalently joins exons to generate mature mRNAs [13]. Conversely, during back-splicing, the downstream 5′ splice site joins backward to the upstream 3′ splice site, resulting in the formation of circular RNAs. This process involves the close proximity of two splicing sites, achieved through at least three different mechanisms: (1) intron pairing facilitated by inverted complementary sequences, (2) involvement of RNA-binding proteins (RBPs), and (3) lariat formation [14,15,16,17,18,19] (Figure 1a,b). 

RNA-seq, followed by bioinformatics analysis, revealed that the existence of complementary *Alu* repeats in long-flanking introns was highly associated with circular RNA formation. Subsequent studies have demonstrated that the formation of circRNAs can be promoted by both repetitive and non-repetitive sequences when these sequences are present within flanking introns [5,14]. 

In addition to cis-acting elements, trans-acting elements such as RBPs also play a regulatory role in circRNA formation. For example, QKI binds to two flanking regions and promotes RNA circularization through dimerization [15]. Other RBPs, including FUS and HNRNPL, influence circRNA biogenesis by binding to introns flanking back-splicing junctions or by stabilizing intronic RNA pairs [17,20,21]. Conversely, RBPs can inhibit circRNA formation by destabilizing double-stranded RNA structures [22,23]. For example, DHX9 is an RNA helicase that unwinds double-stranded RNA regions, preventing the formation of stable secondary structures required for circRNA biogenesis in concert with the RNA-editing enzyme ADAR. 

Exon-containing lariats can serve as circRNA precursors. Large lariats that retain skipped exons can be produced by alternative splicing, which removes exons along with introns. Exon skipping by alternative splicing is proportional to the circularization rate. Exon-containing lariats can undergo internal back-splicing, leading to circRNA formation [19,24,25].

### 2.2. Aberrant Biogenesis of circRNAs in Cancers

Cancer can promote the formation of novel circRNA classes, such as read-through circRNAs (rt-circRNAs) and fusion circRNAs (f-circRNAs) [26] (Figure 1c,d).

The rt-circRNAs originate from read-through transcripts. Read-through transcription occurs when transcription continues through an intergenic region beyond the termination signal, leading to the production of circRNAs from two adjacent genes [27]. Gene pairs generating rt-circRNAs are shorter than random pairs of adjacent genes. rt-circRNAs have characteristics similar to typical circRNAs, including longer introns and more repetitive elements [28]. Read-through circularization can be associated with cancer, where transcription read-through-mediated aberrant gene expression is pervasive [28,29]. Among 460 cancer driver genes, 39 genes were found to produce 67 rt-circRNAs, and the expression of 31 out of 67 was cancer-specific [26]. However, their functional significance in cancer requires further validation.

Cancer-associated chromosomal translocations can lead to the formation of fusion-circRNAs (f-circRNAs). Aberrant chromosomal rearrangements in cancers can result in the juxtaposition of two other wise-separated genes, bringing complementary intronic sequences into close proximity to favor back splicing [30]. In 2016, Guarnerio et al. first showed that f-circRNAs are derived from *PML/RARa* fusion mRNAs in acute promyelocytic leukemia, and that they play role in tumorigenesis independent of their linear transcript and protein counterparts and are implicated in resistance to anti-cancer therapies [30]. Subsequent studies have reported f-circRNAs originated from distinctive chromosomal translocations, including *BCR/ABL1*, *EML4/ALK*, and *SLC34 A12/ROS1* fusions, not only in hematological cancers but also in solid tumors [30,31,32].

Cancer-specific epigenetic modifications, such as promoter CpG island hypermethylation, also play a role in circRNA biogenesis, negatively affecting both circRNA and host gene expression [33].

## 3. Functional Roles of circRNAs in Cancer

CircRNA production is often incompatible with mRNA formation, leading to competition between circRNA production and its corresponding mRNA isoforms, affecting gene expression and protein production [34]. In addition, circRNAs play diverse roles under physiological conditions by modulating gene expression through interactions with microRNAs (miRNAs), RNA binding proteins (RBPs), and chromatin. Some circRNAs serve as templates for translation. Their functional significance has also been highlighted in cancers, where circRNAs have been found to play crucial roles in tumorigenesis, metastasis, and resistance to therapy.

### 3.1. miRNA Sponging

CircRNAs can bind miRNAs and prevent them from interacting with and repressing target RNAs [35]. In 2013, two groups first reported the action of circRNAs as miRNA sponges, exemplified by ciRS-7, which contains more than 79 conserved binding sites for miR-7 [6,9]. miR-7 directly interacts with and downregulates oncogenic genes in cancer-related signaling pathways, and its inhibition by ciRS-7 has been implicated in cancer progression, migration, and invasion of several cancers [10,36,37,38,39,40,41,42]. Subsequent studies have expanded the understanding of circRNAs as miRNA sponges in cancers, identifying circRNAs that regulate oncogenic and tumor-suppressive pathways by inhibiting miRNA activity. One of the most extensively studied circRNAs in cancer is circHIPK3. Zheng et al. profiled circRNAs in cancer tissues compared with those in normal tissues and detected abundant and highly upregulated circHIPK3 expression in cancer tissues. circHIPK3 harbors several binding sites for growth-suppressive miRNAs such as miR-124 and miR-193 [43]. The binding of circHIPK3 to these miRNAs sequesters them, leading to enhanced tumor growth [44]. Conversely, other circRNAs have been found to function as tumor suppressors by sponging oncogenic miRNAs. For example, circITCH has been shown to inhibit proliferation, migration, and invasiveness by regulating the availability of oncogenic miRNAs such as miR-17, miR-214, and miR-224 [45,46]. Likewise, circFOXO3 acts as a sponge for miRNAs such as miR-22, miR-23a, and miR-9, promoting apoptosis and hindering cell cycle progression and metastatic ability [47,48].

### 3.2. Modulation of Protein Function

CircRNAs interact with and regulate the activity of proteins, affecting a wide range of cellular processes critical for cancer development and progression. For example, circFOXO3 functions as a protein scaffold by binding to both MDM2 (an E3 ubiquitin ligase) and the tumor suppressor protein p53, facilitating MDM2-mediated degradation of p53, a key regulator of the cell cycle and apoptosis. The interaction of MDM2 and circFOXO3 also prevents MDM2 from binding to FOXO3, another tumor suppressor, leading to increased FOXO3 levels and tumor apoptosis [49].

Moreover, circRNAs can modulate protein localization and exert tumorigenic functions. CircAMOTL1, which is highly expressed in tumor samples, directly binds to the proto-oncogene c-Myc and promotes its nuclear translocation. This translocation leads to the upregulation of multiple c-Myc targets, supporting tumorigenesis [50].

CircRNAs also affect alternative splicing by interacting with splicing factors. CircRNAs influence the splicing process and gene expression, which often inhibit tumor growth and progression. For example, circUR1 directly interacts with heterogeneous nuclear ribonucleoprotein M (hnRNPM) and modulates alternative splicing of genes involved in cell migration, which negatively impacts cancer metastasis [51]. CircSMARCA5 is another circRNA that interacts with splicing factors to regulate gene expression. It binds to serine–arginine-rich splicing factor 1 (SRSF1) and alters the expression of genes involved in angiogenesis and cell migration, such as vascular endothelial growth factor (VEGF) [52,53].

Certain circRNAs can associate with U1 small nuclear RNA and RNA polymerase II and regulate the transcription of their host genes [54]. Furthermore, circRNAs can modulate host gene transcription by recruiting or decoying proteins. For example, circHuR interacts with CCHC-type zinc finger nucleic acid binding protein (CNBP) and subsequently restrains its binding to *HuR* promoter, leading to repression of HuR expression and tumor progression [55].

### 3.3. Production of circRNA-Derived Peptides or Proteins

CircRNAs are generally expected to be non-coding, but some circRNAs can be translated into peptides or small proteins with functional roles [35]. Naturally occurring circRNAs were first shown to be translated in cap-independent mechanisms by AbouHaidar et al. in 2014 [56]. CircZNF609 is an example of a circRNA that undergoes translation. It was shown that the highly conserved 5′ untranslated region (UTR) of ZNF609 is present in circZNF609 and possesses internal ribosome entry site (IRES)-like activity to facilitate translation [57]. Several other examples associated with cancer pathologies have been identified. For example, circFBXW7 encodes a 21 kDa protein which reduces the half-life of c-Myc by antagonizing the deubiquitinating enzyme USP28. The expression of circFBXW7 is reduced in glioblastoma samples and is positively correlated with glioblastoma patient survival [58]. CircZKSCAN1 can also be translated into a 206aa protein and sensitizes hepatocellular carcinoma cells to the anti-cancer drug sorafenib. Mechanistically, the encoded peptide promotes the interaction between FBXW7 (E3 ubiquitin ligase) and the mammalian target of rapamycin (mTOR) and the degradation of mTOR [59].

Moreover, circRNA-derived peptides can agonize or antagonize cellular signaling associated with proteins encoded by the parent mRNAs, thereby affecting carcinogenesis. For instance, circβ-catenin stabilizes full-length β-catenin by antagonizing GSK3β-mediated phosphorylation and degradation, leading to activation of the Wnt pathway and liver cancer growth [60]. The protein generated by circMAPK1 competes with MEK1 and suppresses the activation of MAPK1 and its downstream signaling [61].

More recently, Huang et al. showed that peptides produced from the tumor-specific circRNA, circFAM53B, stimulate anti-tumor immunity. The authors identified HLA class I binding peptides translated from circFAM53B, which prime T cells in an antigen-specific manner and are associated with the infiltration of antigen-specific CD8^+^ T cells [62].

### 3.4. Emerging Role of circRNA as a Chromosome Translocation Diver

CircRNAs can form specialized structures composed of an RNA–DNA hybrid and displaced single-stranded DNA. R-loops negatively affect transcription, replication, and DNA repair, leading to genome instability [34,63]. Conn et al. found that circR-loops (circRNA: DNA hybrids) are overrepresented in some genes, including *MLL* and *MLLT1–3*, which are involved in leukemogenic chromosomal translocations [64]. CircR-loops are enriched in the *MLL* recombinome and promote transcriptional pausing, DNA breakage, and consequently chromosomal translocation, suggesting that circRNAs can be endogenous RNA carcinogens that drive leukemogenic oncogenes.

## 4. CircRNAs as Biomarkers in Cancers

### 4.1. Diagnostic and Predictive Value of circRNAs in Cancers

CircRNAs have been found to exhibit differential expression profiles in cancers compared to normal tissues. Their dysregulation is implicated in various aspects of tumorigenesis, including cell proliferation, apoptosis, angiogenesis, invasion, and metastasis [35,46,51,52,53,65,66,67,68,69,70]. The clinical utility of circRNAs as biomarkers has been underscored because of their stability in biofluids, such as blood and urine. Numerous studies have explored the potential of circRNAs as biomarkers for cancer diagnosis and prognosis, and these have been summarized in previous reviews [71,72,73,74,75]. Recently, circRNA biomarkers, particularly those involving combinations of multiple circRNAs, have been suggested to enhance diagnostic accuracy. One example is a circRNA classifier that can detect high-grade prostate cancer of grade 2 or greater with five circRNAs (circPDLIM5, circSCAF8, circPLXDC2, circSCAMP1, and circCCNT2) from a urine extracellular vesicle [76]. In addition, Xu et al. (2024) identified five circRNAs (hsa_circ_0060733, hsa_circ_00061117, hsa_circ_0064288, hsa_circ_0007895, and hsa_circ_0007367) from a plasma-based liquid biopsy that can differentiate patients with pancreatic ductal adenocarcinoma (PDAC) between early stage (stage I/II) and late stage (stage III/IV). When combined with cancer antigen 19–9 levels, the diagnostic performance of this circRNA panel for identifying patients with PDAC significantly improved [55]. Fan et al. also demonstrated that the expression of circBRIP1 is associated with the degree of tumor differentiation and the tumor, node, and metastasis (TNM) stage in patients with non-small cell lung cancer (NSCLC) [77]. Plasma circBRIP1 is highly overexpressed in patients, suggesting the possibility of circBRIPP1 serving as a biomarker.

Moreover, circRNAs can serve as predictive markers for treatment response and resistance. Several circRNAs have been identified that modulate chemosensitivity, highlighting the potential of circRNAs in guiding therapeutic strategies [78,79,80,81,82,83,84]. A recent example is the circRNA signature to predict the efficacy of immunotherapy [85]. The ICBcircSig score was established based on the weighted expression of circTMTC3 and circFAM117B from melanoma in cohorts treated with anti-PD-1 or combined anti-CTLA4 and anti-PD1 therapy and showed that the ICBcircSig core model can serve as a robust predictor of the efficacy of immunotherapy in patients with melanoma.

Although accumulating data have demonstrated the relevance of circRNAs in diagnosis and personalized cancer therapy, the clinical translation of circRNAs is still in the early stages. The ongoing development of robust and standardized methods for the detection and quantification of circRNAs will enable their reliable application as biomarkers.

### 4.2. Ongoing Clinical Trials

As of July 2024, five clinical trials on circRNAs have been initiated to evaluate their potential as biomarkers (Table 1). To date, no clinical reports on this topic have been published.

## 5. circRNAs as Therapeutic Platform in Cancer

The key advantages of circRNAs over linear RNAs are their stability and durability, making them promising therapeutic agents. These characteristics allow circRNAs to serve as efficient vectors for protein expression. In 2022, Qu et al. demonstrated that a circRNA encoding the spike protein of SARS-CoV-2 elicited potent neutralizing antibody and T cell responses against SARS-CoV-2, with higher and more durable antigen production than linear mRNA [86]. Recently, the Coalition for Epidemic Preparedness Innovations (CEPI) has collaborated with the Houston Methodist Research Institute to develop circRNA vaccines against COVID-19, tuberculosis, and monkeypox [87]. These efforts represent the therapeutic potential of circRNAs beyond infectious diseases, extending to cancer vaccines aimed at activating host anti-tumor immunity [88]. Cancer vaccines are designed to target mutant proteins and neoantigens specific to tumors. Although current research on circRNAs remains mostly at the preclinical stage, circRNAs may serve as the next generation of cancer therapeutics, offering novel avenues for tackling cancers [89].

### 5.1. In Vitro Synthesis of circRNAs

With the growing hope for circRNAs as promising platforms for therapeutics, various approaches have been developed for the in vitro generation of circRNAs, including ligase-mediated circularization and ribozyme-mediated circularization.

#### 5.1.1. In Vitro Synthesis of RNA Precursor

The generation of circRNAs begins with the synthesis of a linear RNA precursor. The linear RNA precursor can be synthesized chemically or through enzyme-dependent methods. Chemical synthesis enables the production of high-purity RNA precursors with diverse termini and the incorporation of non-natural nucleotides. However, it is generally limited to the synthesis of short oligomers, approximately 50~70 nucleotides in length [90,91].

More commonly, RNA precursors are generated by in vitro transcription using DNA-dependent RNA polymerases, typically derived from bacteriophages such as T7 or SP6. This approach facilitates the preparation of much longer RNA molecules with higher yields compared to chemical synthesis [92]. However, a limitation of this method is the generation of heterogeneous products, which may include RNA molecules that are either shorter or longer than desired [93]. Addressing this variability is crucial for improving the reliability and efficiency of precursor RNA synthesis.

#### 5.1.2. Ligase-Mediated Circularization

End ligation of linear RNAs is facilitated by T4 DNA and RNA ligases derived from T4 bacteriophages (Figure 2a). T4 DNA ligase has nick-closing activity in double-stranded DNA as well as ligation activity at the two DNA ends [94]. An auxiliary oligonucleotide (splint) complementary to the terminus of the RNA precursor is used to juxtapose the free end of the single-stranded RNA, allowing nick formation [94,95]. T4 DNA ligase has a low tolerance for mismatches and therefore results in RNA ligation with high efficiency and specificity [92].

T4 RNA ligase I catalyzes the formation of 3′, 5′ phosphodiester bonds in single-stranded RNA. The free ends of single-stranded RNA should be in close proximity; otherwise, RNA circularization can compete with intermolecular ligation. Similar to the T4 DNA ligase reaction, helper DNA or splints can be used to assist in ligation by orienting the free ends of RNA substrates properly and preventing the substrate from folding into an unsuitable structure [96,97].

T4 RNA ligase II works on both single- and double-stranded RNAs, but more efficiently on nicks in double-stranded RNA. Appropriate ligation using T4 RNA ligase I requires a splint. Recently, Chen et al. (2020) reported a high-yield circularization strategy that did not require a splint (terminal base-pair strategy). This strategy is based on secondary structure prediction, which allows for the design of precursor RNA to form a few base pairs at the ligation site [98].

#### 5.1.3. Ribozyme-Mediated Circularization

In addition to ligation-mediated circularization, a spontaneous self-splicing system has been used to produce circular RNA (Figure 2b). Introns are classified into four main categories, based on their structure and excision mechanisms: spliceosomal, group I splicing, group II splicing, and tRNA introns [99]. Group I and group II introns are RNA enzymes, ribozymes, that catalyze their splicing [100].

Group I introns are found in rRNA and tRNA in bacteria and non-metazoan eukaryotes and undergo splicing without assistance of proteins [92,99]. Since the function of splice sites is not dependent on their relative positions (5′ or 3′), but on sequences and structural features, it is possible to reverse the order of the splice sites to autocatalytically generate RNA circles using permuted forms of group I introns [101]. Mechanistically, this method involves (1) the attack of the 3′ hydroxyl group of guanosine (G) on the 5′ splice site, releasing the 3′ terminal sequence (5′ half intron) and (2) the attacks of the newly generated free 3′ hydroxyl group on the 3′ splice site, releasing the 3′ half intron [97,101]. This method has been reported to be more applicable to the circularization of long RNAs [102].

For RNA circularization in vitro, the pre-tRNA^leu^ gene of *Anabaena* and the thymidylate synthase gene of the T4 bacteriophage were used to generate a permuted intron-exon (PIE) construct. In the PIE construct, the 5′ segment of the group I intron is relocated to the tail of the exon, and the remaining 3′ segment of the intron is placed at the head of the same exon [92]. Wesselhoeft et al. demonstrated that this strategy allows for the successful RNA circularization of a wide range of RNAs up to 5 kb in length in vitro through self-splicing [102]. However, it can result in the undesired retention of small terminal fragments of exogenous exons at the end-joining site, referred to as a scar.

Group II introns have been mostly found in protein-coding genes. As in nuclear mRNA introns, they initiate splicing by attacking the 3′ hydroxyl group within introns on the 5′ splice site, forming an intron lariat in which the 5′ end of the intron is linked by a 2′-5′ phosphodiester bond near the 3′ end of intron [100]. Initially, yeast mitochondrial genes were used for RNA circularization [103,104]. Recently, Chen et al. (2022) have adopted the group II self-splicing intron from the surface layer protein of *Clostridium tetani* to generate “scarless” circRNAs. The authors introduced modifications to the exon binding sites within the intron, enabling the formation of base pairs with the 3′ and 5′ ends of the circular exon without extraneous sequences [105].

### 5.2. Engineering Circular RNA Translation

Several studies have confirmed that circRNAs bind to polysomes and produce proteins mediated by the IRES or short sequences containing m6A modifications [57,106,107]. Owing to their ability to sustain durable protein production, circRNAs have emerged as promising vectors for protein expression. Continuous efforts are being made to engineer circRNAs for efficient cellular translation.

#### 5.2.1. IRES-Mediated Translation

IRES are specialized RNA elements that recruit small ribosomal subunits to the internal positions of mRNA and are commonly found in viral RNAs [108]. Recently, Chen et al. (2021) systematically examined a library of 5,500 synthetic oligos derived from viral and human genes to identify sequences that facilitate circRNA translation [109]. They found that IRES activity on circRNAs did not correlate with their activity on linear RNAs and appeared to depend more on structural features and complementarity to 18s rRNA. However, the oligos in this study were limited to ~200nt length sequences, which may not capture the impact of longer IRES elements. In a follow-up study, a modular circRNA assembly platform was used to systematically examine the effects of factors such as vector topology, UTR, and IRES on protein production [110]. This study concluded that full-length IRES, particularly those from the type I class (which contain complex secondary structures and require eukaryotic initiation factors [111]), exhibit stronger translation activity in the context of circRNAs.

UTRs harbor important regulatory elements that affect translation by interacting with various RNA-binding proteins [112]. Besides IRES, the insertion of specific 5′ and 3′ UTR sequences can positively affect circRNAs translation. For example, incorporating PABP motifs and the eIF4G recruiting aptamer in the 5′UTR and HBA1 sequence in the 3′ UTR significantly improved translation efficiency [110].

Ongoing efforts aim to identify optimal UTR combinations for linear RNAs [113,114] and comprehensive studies on various UTR combinations in circRNAs are necessary to further enhance their translational activity.

#### 5.2.2. M6A-Mediated Translation

N6-mehtyladenosien (m6A) is the most abundant base modification in RNAs and play diverse roles in translation. Specifically, m6A in the 5′ UTR has been shown to facilitates cap-independent translation during heat shock stress, while m6A in 3′ UTR promotes translation via its binding protein YTHDF1 [115,116].

Yang et al. found that circRNA translation can be achieved through sequences including HRRACH motifs, which resemble the target motifs of m6A modification. The authors demonstrated that several circRNAs contain extensive m6A modifications, which drive protein translation via the m6A reader YTHDF3 and translation initiation factors [106]. Chen et al. (2023) examined the effect of m6A incorporation on circRNA synthesis. Their study showed that m6A incorporation does not adversely affect circRNA translation, and rather confers improved stability [110].

### 5.3. Applications in Cancer Therapy

The application of circRNAs in cancer therapy is multifaceted. They can be used as cancer immunogens and adjuvants. A recent proof-of-concept study showed the robust cytotoxic T cell-mediated anti-tumor activity of a circRNA-based vaccine for melanoma by generating the shared antigen ovalbumin (OVA). OVA-coding circRNAs encapsulated in lipid nanoparticles (LNP) were intramuscularly administered to tumor-bearing mice and showed excellent results in treating different types of tumors, including colorectal cell tumors, orthotopic melanoma, and lung metastasis models [117]. Another study demonstrated the potential of circRNAs as antigens in an active immune system. Xu et al. (2022) investigated a circRNA vaccine using circRNAs extracted from colon cancer cells by injecting them into subcutaneous tumor-bearing mice. They observed that circRNAs can induce innate and adaptive immunity and cancer regression [118] (Figure 3a).

Moreover, circRNAs can act, not only as immunogens, but also as vaccine adjuvants. Several groups have suggested the potential role of circRNAs as modulators of the immune system. Chen et al. (2017) demonstrated that exogenous circRNAs trigger an immune response via the nucleic acid sensor RIG-I, independently of a 5′ triphosphate, double-stranded RNA structure or the primary sequence [119]. The authors suggested that introns were the major determinants of immunogenicity. They demonstrated the robust immunogenicity of RNA circularization by PIE splicing using the group I intron of the thymidylate synthase gene (T4 bacteriophage) compared to direct back-splicing using the endogenous human ZKSCAN1 intron. In a subsequent study, the immunogenicity of circRNAs was shown to be modulated by m6A modification [120]. Liu et al. found that circRNAs generated by a permuted intron–exon splicing strategy using a group I intron induced PKR activation, but that circularized RNA produced by direct ligation with T4 RNA ligase did not significantly trigger PKR activation [121].

Recently, Amaya et al. systematically examined the adjuvant activity and antigen-specific immunity of circRNA vaccines in mice. They observed that when combined with OVA protein, circRNAs induced T cell responses similar to those produced by common vaccine adjuvants (AddaVax) with OVA protein in the lung, spleen, and lymph nodes after subcutaneous injection. Additionally, a stronger T cell response was observed in the lung when the circRNAs were inoculated intranasally. In the same study, the authors showed that circOVA, delivered intraperitoneally in a complex with CART (charge-altering releasable transporter), eradicated OVA-expressing subcutaneous tumors, demonstrating that circRNAs serve as effective cancer immunotherapies as adjuvants and immunogens [122] (Figure 3b).

**Figure 3 ijms-25-10121-f003:**
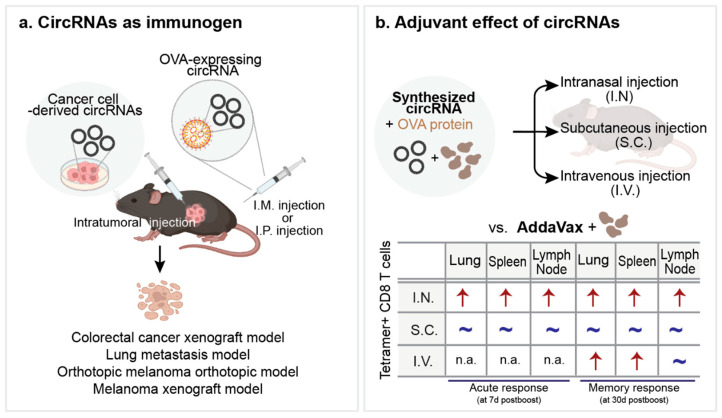
Applications of circRNAs in cancer therapy. (**a**) circRNAs as immunogen. Antigen-expressing circRNAs or cancer cell-derived circRNAs have been used to induce an immune response against tumors. In vitro synthesized circRNAs encapsulated with carriers (e.g., LNP or CART) were administered intramuscularly (I.M. injection) or intraperitoneally (I.P. injection). Cancer-cell derived circRNAs were administrated via intra-tumoral injection. (**b**) Summary of adjuvant activity of circRNAs administered through three different routes, as determined by tetramer + CD8 T cells. Red arrow: significantly robust induction as compared to AddaVax. ~: comparable induction to AddaVax. ↑: Stronger T cell response. n.a.: not available [122].

Given the versatility of circRNAs in modulating the immune response, circRNA-based therapies hold promise for advancing personalized cancer treatment. They can be tailored to target specific tumor antigens that align with the unique profiles of individual cancers. With ongoing efforts to optimize the immunogenic properties of circRNAs, circRNA-based therapies could be combined with other immunotherapies or conventional cancer therapies to develop more comprehensive and effective cancer treatment strategies.

## 6. Conclusions and Future Perspective

CircRNAs have become pivotal players in the field of RNA biology and therapeutics. Accumulating evidence has demonstrated that their dysregulated expression and aberrant biogenesis can be implicated in cancer, underscoring their potential as biomarkers for cancer diagnosis and therapeutics. Their stable and durable characteristics enable non-invasive, liquid biopsy-based assays with circRNAs, facilitating early detection of diseases, including cancer, and personalized treatment courses. To apply circRNA therapeutics into clinical practice, more sensitive and specific detection and standardized quantification methods are necessary.

The promising applications of circRNAs extend beyond biomarkers to therapeutic platforms. Recent advances in circRNA synthesis and translation efficiency have made circRNAs emerging candidates for nucleic acid therapeutics. Ongoing research on their molecular mechanisms in cancer and the continuous exploration of circRNA engineering will enable the integration of circRNAs into clinical practice in the near future.

## Figures and Tables

**Figure 1 ijms-25-10121-f001:**
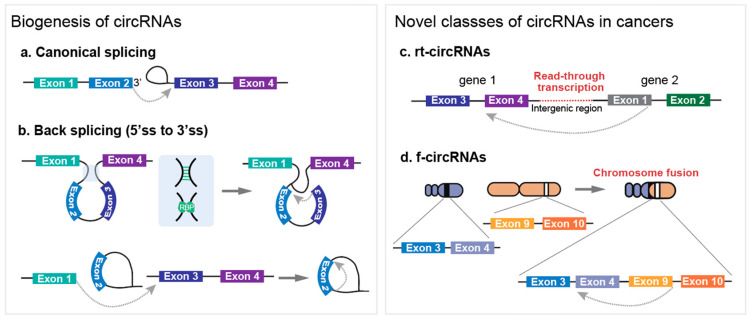
Biogenesis of circRNAs and novel classes of circRNAs in cancers. (**a**) Canonical splicing. (**b**) Main mechanisms of back-splicing: (upper panel) intron-pairing facilitated by inverted complementary sequences and the RBPs; (bottom panel) lariat formation. (**c**) Generation of rt-circRNAs from read-through transcripts. (**d**) Generation of f-circRNAs by chromosome fusions.

**Figure 2 ijms-25-10121-f002:**
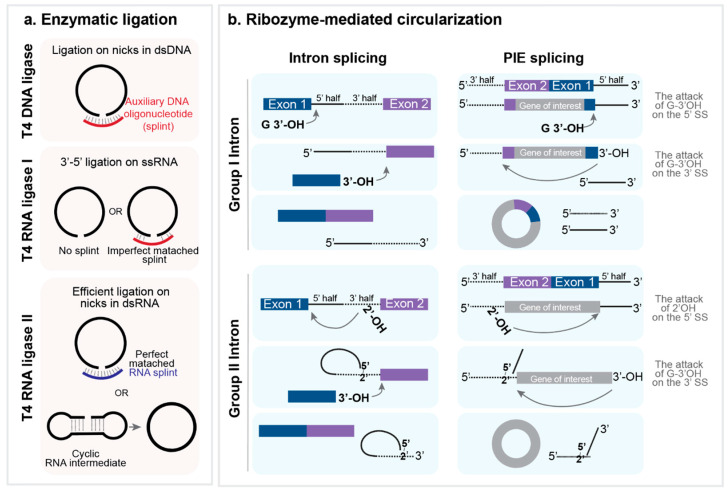
In vitro circRNA synthesis. (**a**) Enzyme-mediated RNA circularization. T4 DNA ligase, T4 RNA ligase I, and T4 RNA ligase II facilitate end ligation of linear RNA precursor in concert with auxiliary DNA or RNA oligonucleotides, leading to circRNA formation. (**b**) Ribozyme-mediated RNA circularization. circRNAs generation can be achieved by exon intron permutation methods.

**Table 1 ijms-25-10121-t001:** Ongoing clinical trials investigating circRNAs as biomarkers in various cancers.

Study	Cancer	circRNAs	Objective	Sponsor
NCT 05934045	ALK positive anaplastic large-cell lymphoma	circRNAs in patient blood	Prognostic and/or predictive markers/therapeutic targets	University Hospital, Toulouse
NCT 04464122	Neuroendocrine Neoplasm	circRNAs in patient platelets	Diagnostic and predictive markers	University of Roma La Sapienza
NCT 06042842	Hepatocellular carcinoma	has_circ_0004001 in patient blood	Diagnostic marker for early stages of disease	Assiut University
NCT 05771337	Breast cancer	has_circ_0001785(circELP3) has_circ_100219(circFAF1) in patient blood	Diagnostic and prognostic biomarker	Assiut University
NCT 03334708	Pancreatic adenocarcinoma	circRNAs in patient blood	Diagnostic marker for early stages of disease and predictive marker	Memorial Sloan Kettering Cancer Center
NCT 04584996	Pancreaticobiliary cancer	circRNAs in patient blood	Diagnostic marker	Royal Surrey County Hospital NHS Foundation Trust

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
