# Peer review of "Circular RNAs: Novel Players in Cancer Mechanisms and Therapeutic Strategies"

_ijms, 2024, doi:10.3390/ijms251810121_

Round 1

Reviewer 1 Report

Comments and Suggestions for Authors

Dear Author,

The manuscript is very well written, interesting, and comprehensive. It introduces in the field of circRNA biology and provides information on their potential application in diagnostics and therapeutics, including the latest research and data. I just have a few minor comments that are not associated with the overall quality of the review:

Gene names should be written in italics (e.g., line 109).

Figure 1b: I suggest denoting single mechanisms in this figure since just two mechanisms can be easily distinguished.

Line 133: "can bind miRNAs"

Line 202: The 4.1 subheading title is not clear and grammatically correct.

Table 1 has no title and caption. Moreover, instead of "Has_circ," it should be "hsa_circ", and instead of "presitive markers" should be "predictive markers" (I suppose).

Line 287: The 5.1.2 subheading title is not clear because in line 263 such a mechanism is not mentioned. Is it equivalent to PIE splicing strategy?

Line 315: Latin names should be in italics.

Best wishes

Author Response

We thank the reviewer for the comments and for finding our manuscript interesting and comprehensive. We have revised our manuscript according to the reviewer’s comments, which have improved the quality of the manuscript.

Gene names should be written in italics (e.g., line 109).

Line 315: Latin names should be in italics.

Response : We have revised the text to ensure that gene names and Latin names are italicized.

Figure 1b: I suggest denoting single mechanisms in this figure since just two mechanisms can be easily distinguished.

Response : We have revised the caption from “three main mechanisms” to “main mechanisms” as we agree that just two mechanisms can be easily distinguished. 

Line 133: "can bind miRNAs"

Line 202: The 4.1 subheading title is not clear and grammatically correct.

Table 1 has no title and caption. Moreover, instead of "Has_circ," it should be "hsa_circ", and instead of "presitive markers" should be "predictive markers" (I suppose).

Response : We apologize for the errors. We have corrected the typos and have added the title for Table 1.

Line 287: The 5.1.2 subheading title is not clear because in line 263 such a mechanism is not mentioned. Is it equivalent to PIE splicing strategy?

Response : We thank the reviewer for pointing this out. While the PIE splicing strategy involves Group I and II introns, which are ribozymes as mentioned in line 328, we agree that the subheading may cause confusion for general readers. To address this, we have revised the paragraphs under section 5.1 and 5.1.3 to improve clarity and coherence. Specifically, we replaced the phrase “enzyme-dependent end-ligation of linear RNA and the permuted intron-exon (PIE) splicing strategy” with “ligase-mediated circularization and ribozyme-mediated circularization” to ensure better understanding.

Reviewer 2 Report

Comments and Suggestions for Authors

The author presents a comprehensive overview of the current state of knowledge regarding circRNA biogenesis, delving into their intricate mechanisms of action in cancer, and appraises their promising clinical potential as biomarkers. Furthermore, a potential therapeutic strategy leveraging circRNAs is thoughtfully explored. I have some comments to the author:

Comments:

1.      The section of miRNA sponging and Modulation of protein function are too simply. It should be included more special roles circRNAs in cancer research.

2.      There can be provide a table about the circRNA as biomarkers in cancers in 4.1 section.

3.      In the ‘circRNAs as therapeutic platform in cancer’ section, it would be invaluable to discuss the diverse methods employed for synthesizing circRNAs, including their respective strengths and weaknesses.

4.      In the ‘Applications in cancer therapy’ section, the author should point the application prospects and directions of the circRNA in cancer therapy.

5.      Some references missed the page number.

6.      A thorough proofreading pass to identify and correct any typographical errors is essential.

Comments on the Quality of English Language

Please see the comments.

Author Response

We thank the reviewer for the insightful suggestions. We have revised our manuscript according to the reviewer’s comments, which have significantly improved the quality of the manuscript.

Comments:

  1. The section of miRNA sponging and Modulation of protein function are too simply. It should be included more special roles circRNAs in cancer research.

Response : In response to the reviewer’s concern, we have expanded Sections 3.1 and 3.2 to provide more detailed information on the roles of circRNAs in miRNA sponging and the modulation of protein function in cancer research. In particular, we have added further discussion on the transcriptional regulation by circRNAs through their interactions with proteins in Section 3.2. We believe that these revisions will enhance the readers’ comprehension of the topic. We would be happy to be guided by the reviewer if additional modifications are necessary.  

  1. There can be provide a table about the circRNA as biomarkers in cancers in 4.1 section.

Response : We appreciate the reviewer’s suggestion and acknowledge that the number of circRNAs being identified as biomarkers is rapidly growing. To avoid including lengthy and exhaustive information, which has been covered in previous review articles, we have focused on presenting the most recently reported biomarkers, particularly highlighting those that involve combinations of multiple circRNAs for improved diagnostic, prognostic, and predictive value. We have revised the manuscript to make this focus clearer. If the reviewer feels that a table summarizing the biomarkers discussed in this manuscript would be helpful, we would be happy to include it. 

  1. In the ‘circRNAs as therapeutic platform in cancer’ section, it would be invaluable to discuss the diverse methods employed for synthesizing circRNAs, including their respective strengths and weaknesses.

Response : As suggested by the reviewer, we have devoted a paragraph to discussing the methods for synthesizing RNA precursors with advantages and challenges associated with each method. This new section can be found under the subheading  5.1.1 In vitro synthesis of RNA precursor in the revised manuscript. 

  1. In the ‘Applications in cancer therapy’ section, the author should point the application prospects and directions of the circRNA in cancer therapy.

Response : In response to  reviewer’s comment, we have revised the ‘Applications in cancer therapy’ section to include a discussion on the prospects and direction of circRNA-based therapies in cancer.

  1. Some references missed the page number.

Response : We appreciate the reviewer’s attention to detail. While we have used EndNote to manage and cite references, we acknowledge the importance of accuracy. As manually adding page numbers to each reference is challenging, we have ensured that DOI numbers are provided for the references lacking page numbers.

  1. A thorough proofreading pass to identify and correct any typographical errors is essential.

Response : We apologize for the errors. We have carefully reviewed the manuscript and have corrected all identified typos.